# Impact of Three Different Processing Techniques on the Strength and Structure of Juvenile Ovine Pulmonary Homografts

**DOI:** 10.3390/polym14153036

**Published:** 2022-07-27

**Authors:** Johannes J van den Heever, Christiaan J Jordaan, Angélique Lewies, Jacqueline Goedhals, Dreyer Bester, Lezelle Botes, Pascal M Dohmen, Francis E Smit

**Affiliations:** 1Department of Cardiothoracic Surgery, Faculty of Health Sciences, University of the Free State (UFS), P.O. Box 339 (Internal Box G32), Bloemfontein 9300, South Africa; jordaancj@ufs.ac.za (C.J.J.); lewiesa@ufs.ac.za (A.L.); besterd@ufs.ac.za (D.B.); pascal.dohmen@med.uni-rostock.de (P.M.D.); smitfe@ufs.ac.za (F.E.S.); 2Department of Anatomical Pathology, Faculty of Health Sciences, University of the Free State (UFS), P.O. Box 339 (Internal Box G32), Bloemfontein 9300, South Africa; goedhalsj@ufs.ac.za; 3Department of Health Sciences, Central University of Technology, Free State (CUT), Private Bag X20539, P.O. Box 339 (Internal Box G32), Bloemfontein 9300, South Africa; botesl@cut.ac.za; 4Klinikdirektor (k), Klinik und Poliklinik für Herzchirurgie, Universitätsmedizin Rostock, Schillingallee 35, 18057 Rostock, Germany

**Keywords:** homografts, ischaemic harvesting, decellularization, cryopreservation, glutaraldehyde-fixation

## Abstract

Homografts are routinely stored by cryopreservation; however, donor cells and remnants contribute to immunogenicity. Although decellularization strategies can address immunogenicity, additional fixation might be required to maintain strength. This study investigated the effect of cryopreservation, decellularization, and decellularization with additional glutaraldhyde fixation on the strength and structure of ovine pulmonary homografts harvested 48 h post-mortem. Cells and cellular remnants were present for the cryopreserved group, while the decellularized groups were acellular. The decellularized group had large interfibrillar spaces in the extracellular matrix with uniform collagen distribution, while the additional fixation led to the collagen network becoming dense and compacted. The collagen of the cryopreserved group was collapsed and appeared disrupted and fractured. There were no significant differences in strength and elasticity between the groups. Compared to cryopreservation, decellularization without fixation can be considered an alternative processing technique to maintain a well-organized collagen matrix and tissue strength of homografts.

## 1. Introduction

End-stage heart valve disease mandates the repair or replacement of a patient’s diseased heart valve/s with either mechanical or biological valve prostheses. Mechanical prostheses demonstrate superior durability and longevity in patients; however, recipients require lifelong anticoagulation therapy. In contrast, bioprosthetic valves (including glutaraldehyde (GA)-fixed porcine valves or bovine pericardium mounted onto a frame, free xenograft valves, or donor homograft valves) do not require continuous anticoagulation therapy but have limited durability and require more frequent reoperation [1]. Currently, cryopreserved pulmonary homografts remain the valve of choice for the replacement of the native pulmonary valve in the Ross procedure [2], as well as for the reconstruction of the right ventricle outflow tract (RVOT) in children with congenital abnormalities [3]. Unfortunately, the early degeneration of these homografts occurs in younger patients [4], and there is a lack of availability, especially for smaller-sized conduits suitable for neonates [5].

According to the internationally accepted guidelines, homografts from either beating or non-beating heart donors should be harvested and processed within 24 h after death to retain maximum cell viability. This guideline restricts the available post-mortem donor pool significantly. Various efforts have been made to address the shortages in homograft availability, and our research group has proven that the post-mortem ischaemic time can be extended safely to around 48 h without affecting valve performance [6]. Although cryopreservation is currently the most frequently used and probably the best method for long-term storage of homografts [7], it does damage the collagen scaffold, irrespective of ischaemic harvesting time [8]. The effect of cryopreservation on the collagen scaffold might be of greater importance in determining the long-term survival of the homograft than the impact of extending the post-mortem ischaemic time prior to harvesting. The presence of donor cell antigens in cryopreserved homografts is also associated with an adverse immunological response from the recipient, resulting in valve calcification and degeneration [9,10,11].

Decellularization processes remove interstitial and endothelial cellular components from a homograft valve and can potentially lead to the creation of a valve with significantly reduced immunogenicity and reduced calcification [12]. There are concerns about maintaining the strength of the conduit once the cellular components are removed [13]. Furthermore, using multi-detergent, enzyme-based decellularization methods might also decrease flexural stiffness, disrupt the extracellular matrix (ECM) structure [14] and reduce the glycosaminoglycan (GAG) content [15]. These GAGs fill most of the extracellular space and provide mechanical support to the tissue [16]. Therefore, the additional fixation and stabilization of the collagen scaffold with GA might be required [17]. However, the free aldehyde groups of GA are associated with cellular toxicity [18], while conjugated GA may contribute to graft calcification [19,20]. Washing and detoxifying with solutions that can bind the free aldehyde groups prior to implantation can improve durability and biocompatibility [21]. EnCap technology describes the fixation of biological tissue with GA and the subsequent treatment of the GA-fixated tissue with a high concentration liquid polyol like propylene glycol (PG). The PG binds to the free aldehyde groups of the GA, blocking the free aldehyde groups as a potential binding site for calcium and other minerals and thereby mitigating the calcification of the tissue [22]. From a study evaluating the performance of commercially stent-mounted GA-preserved aortic homografts (not decellularized), it was concluded that better long-term performance of the homograft tissue compared to GA-fixed heterologous biological valves could be expected [23]; however, other studies using GA-fixed decellularized homograft valves are lacking.

The aim of this study was to compare the morphology and mechanical properties of standard cryopreserved pulmonary homografts to decellularized pulmonary homografts and decellularized pulmonary homografts treated with EnCap™ AC technology, evaluating the impact of the processing method on ovine homografts harvested after 48 h cold ischaemia. Our proprietary decellularization protocol with proven synergy was used [24].

## 2. Materials and Methods

### 2.1. Study Design

Pulmonary homografts harvested from juvenile Dorper sheep (*n* = 15), with a post-mortem cold ischaemic harvesting time of 48 h, were divided into three groups: Group 1, cryopreserved (*n* = 5); Group 2, decellularized (*n* = 5); and Group 3, decellularized plus EnCap^TM^-treated (*n* = 5). After processing, the structural integrity, strength, and elasticity of the leaflet and wall tissue were evaluated and compared. Strength and elasticity analysis included tensile strength (TS) and Young’s modulus (YM), and morphological evaluation included DAPI staining to confirm decellularization, hematoxylin and eosin staining (H&E), von Kossa staining, modified Verhoeff’s van Gieson staining, scanning electron microscopy (SEM), and transmission electron microscopy (TEM). A schematic representation of the study design is given in Figure 1. The interfaculty Animal Ethics Committee of the University of the Free State (UFS-AED2016/0101) approved the study.

### 2.2. Preparation of Homografts

Heart-lung blocks (*n* = 15) from freshly slaughtered juvenile Dorper sheep with a bodyweight of 24–30 kg were collected from a local abattoir and subjected to 48 h ischaemia at 4 °C before dissection and processing, as previously described by Bester et al. [15]. Pulmonary homografts (*n* = 15) were harvested and washed in copious amounts of cold (4 °C) Ringers lactate solution (Fresenius Kabi/Intramed, Midrand, South Africa). All homografts were sterilized overnight at 4 °C in 100 mL Medium199 (Whitehead Scientific, Johannesburg, South Africa) and an antibiotic cocktail consisting of 2.5 mg Amphotericin B (Bristol-Myers Squibb, Bedfordview, South Africa), 50 mg Piperacillin (Brimpharm South Africa (Pty) Ltd., Cape Town, South Africa ), 50 mg Vancomycin (Gulf Drug Company, Mount Edgecombe, South Africa), and 25 mg Amikacin sulphate (Bodene (Pty) Ltd., trading as Intramed, Port Elizabeth, South Africa). Valves in Group 1 (*n* = 5) were cryopreserved the next day in 100mL Medium 199 + 11mL dimethyl sulfoxide (DMSO) (Highveld Biological, Johannesburg, South Africa) in a Cryoson controlled rate freezer (Consarctic, Schöllkrippen, Germany) at a rate of −1 °C/min to −140 °C and stored in the vapor phase of liquid nitrogen (LN_2_) until evaluation.

Decellularized homografts (Group 2, *n* = 5) were prepared according to our proprietary protocol [24,25] with the addition of Benzonase^®^ (Thermo Fisher Scientific, Johannesburg, South Africa ). Briefly, homografts were subjected to osmotic shock in two changes of hypertonic 5% NaCl-solution and distilled water, repeated changes in a multi-detergent solution (0.5% Sodium deoxycholate (SDS) (Sigma-Aldrich, Johannesburg, South Africa)), 1% Sodium deoxycholate (SDC) (Sigma-Aldrich, Johannesburg, South Africa), and 1% Triton-X 100 (Sigma-Aldrich, Johannesburg, South Africa)] in PBS, followed by numerous washings in PBS, and a half-strength antibiotic cocktail, which consisted of the same antibiotic mixture that was used for the sterilization of cryopreserved valves, under constant shaking. These steps were followed by enzymatic treatment with descending concentrations of Benzonase^®^ [26], repeated washing in PBS, delipidation in 70% ethanol, and final storage in PBS with antibiotics.

Additionally, decellularized homografts (Group 3, *n* = 5) were treated with EnCap^TM^ AC technology and stored in propylene oxide (Sigma-Aldrich, Johannesburg, South Africa) [22]. EnCap^TM^ AC technology includes the GA-tanning of the collagen scaffold, and the binding of a polyol, namely, propylene glycol (Sigma-Aldrich, Johannesburg, South Africa) to the free aldehyde groups to lower toxicity, reducing the host inflammatory response and mitigating calcification.

All homografts were confirmed to be culture-negative. To confirm the effective decellularization of the homografts in Groups 2 and 3, DNA quantification was done with 4′, 6-diamidino-2-fenielindol (DAPI) staining, gel electrophoresis, and NanoDrop counts by the Cardiovascular Research Unit, UCT Medical School [27].

### 2.3. Evaluation of Processed Homografts

#### 2.3.1. Histological Analysis

Leaflet and wall-tissue samples were collected in 4% buffered Formalin, embedded in paraffin wax, sectioned and stained according to standard H&E, von Kossa, and modified van Gieson staining protocols for histological evaluation [28].

Pulmonary leaflet and wall-tissue samples for scanning and transmission electron microscopy were collected in 3% GA and processed according to standard protocols for SEM and TEM evaluations [29]. Tissue specimens for SEM were dried using the critical point method (Tousimis critical point dryer, Rockville, MD, USA, ethanol dehydration, carbon dioxide drying gas) and sputter-coated with gold (BIO-RAD, Microscience Division Coating System, London, UK; Au/Ar sputter coating @ 50–60 nm). A Shimadzu SSX 550 scanning electron microscope (Kyoto, Japan, with integral imaging (SDF, TIF and JPG format)) was used to examine and photograph the tissue surface.

Leaflet and wall samples for TEM were fixed in 3% GA overnight, post fixated in Palade’s osmium tetroxide, and dehydrated in a graded acetone series. Dehydrated samples were impregnated/embedded in an epoxy [29] to facilitate the creation of ultra-thin sections for the TEM evaluation. Ultra-thin sections were cut from the sample embedded in the epoxy using a Leica ultra-microtome (Leica Ultracut UC7, Vienna, Austria). After sectioning, the samples were stained with uranyl acetate and lead citrate. Sections of the leaflet samples were evaluated by using a Philips (FEI, Eindhoven, The Netherlands) CM100 transmission electron microscope and photographed using an Olympus Soft Imaging System Megaview III digital camera, with Soft Imaging System digital image analysis and documentation software (Olympus, Tokyo, Japan).

#### 2.3.2. Mechanical Properties

The mechanical properties of pulmonary valve leaflet and wall samples were examined using a tensile strength testing apparatus (Lloyds LS100 Plus, IMP, Johannesburg, South Africa), according to the method described by Thubrikar et al. [30]. Tissue samples measured 5 × 10 mm, with leaflets and pulmonary sinus wall samples cut in the circumferential direction. These tissue samples were fixed between clamps at both ends and gradually stretched (0.1 mm/s) by applying constant tension on the two ends, and the data was recorded on a personal computer.

#### 2.3.3. Statistical Analysis

All values are expressed as median values with a corresponding range (first and third quartiles). Statistical analyses were performed using GraphPad Prism version 8.3.1 (GraphPad Software, La Jolla, CA, USA, www.graphpad.com). The overall difference between the groups was assessed using a Kruskal–Wallis (KW) test. The significance was set as *p* < 0.05.

## 3. Results

The successful decellularization of both the valve leaflets and the pulmonary artery wall tissue in the decellularized (Figure 2C,D) and decellularized plus EnCap^TM^-treated (Figure 2E,F) groups was shown with DAPI staining.

Gel electrophoresis (fragmented DNA bands < 200 bp) and nanodrop readings of below 50 ng/mg for the same tissue samples were used to confirm and support the successful decellularization. The NanoDrop reading for fresh homograft tissue prior to decellularization was 204.8 ng/mg [200.8–212], and following decellularization no DNA was detectable via NanoDrop. The absence of DNA following the decellularization of homograft tissue was also shown with gel electrophoresis (Figure 3).

H&E staining of leaflet samples from cryopreserved homografts after 48 h of ischaemia demonstrated the presence of an endothelial layer and the uniform distribution of donor interstitial cells in the ECM (Figure 4A). The leaflets of valves in the decellularized and decellularized plus EnCap^TM^-treated groups demonstrated well-preserved collagen matrices without any endothelial or interstitial cells (Figure 4D,G). No calcific deposits are visible in any of the samples on von Kossa staining (Figure 4B,E,H), while large numbers of elastin fibers (black arrows) are clearly demonstrated in the ventricularis region of the leaflets in all three groups on modified van Gieson staining (Figure 4C,F,I). The collagen in the cryopreserved group appeared more collapsed compared to fresh untreated tissue [8], and compared to the cryopreserved group, more loosely arranged in the decellularized group and more compact and dense in the decellularized plus EnCap^TM^-treated group.

Similar results were demonstrated for the wall tissue as for the leaflets, with an endothelial cell layer (white arrow) and uniform donor interstitial cell distribution in the cryopreserved group (Figure 5A) and no cells present in the decellularized scaffold or decellularized plus EnCap^TM^-treated group (Figure 5D,G) on H&E staining. No calcific deposits could be demonstrated with von Kossa staining in any of the three groups (Figure 5B,E,H), and the dense distribution of elastin fibers (black arrows) was demonstrated in all three groups on Modified van Gieson staining (Figure 5C,F,I). Collagen in the walls of the cryopreserved group is again collapsed, loosely arranged in the decellularized group, and dense and compacted in the decellularized plus EnCap^TM^-treated group.

SEM demonstrated cell dehiscence, although it did demonstrate the uniform coverage of leaflets and wall tissue with endothelial cells in the cryopreserved group (Figure 6A,B). Leaflets and walls in both the decellularized (Figure 6C,D) and decellularized plus EnCap^TM^-treated groups (Figure 6E,F) demonstrated exposed collagen networks without any endothelial cells and only basal membrane remnants (white arrows).

TEM demonstrated the presence of cells and cellular remnants in the cryopreserved group (Figure 7A,B), and no interstitial cells were present in the leaflets and wall tissue in the decellularized group (Figure 7C,D) or decellularized plus EnCap^TM^-treated group (Figure 7E,F). Collagen bundles appeared more loosely arranged in the decellularized group (Figure 7C,D) and more compressed and dense in the decellularized plus EnCap^TM^-treated group (Figure 7E,F).

The Young’s modulus of the decellularized leaflets suggests more extensible leaflet tissue, but no statistically significant differences in the tensile strength or Young’s modulus of the leaflet or wall tissue (Table 1) between the three groups could be demonstrated. The stiffness of the leaflets in the decellularized plus EnCap^TM^-treated group did increase when compared to that in the other two groups; however, the increase was not significant (*p* > 0.05).

## 4. Discussions

Ross (1965) and Barratt-Boyes et al. (1965) were the first to describe the successful use of “fresh” or homovital homografts in the aortic position and pulmonary homografts for RVOT reconstruction, with acceptable homograft performance [31,32]. The superior performance and increased long-term durability of cryopreserved homografts with maintained cellular viability, compared to valves stored at 4 °C [33,34,35], unfortunately, led to an internationally accepted guideline that homografts from either beating heart or non-beating-heart donors should be harvested and processed within 24 h after death to retain maximum cell viability. These findings not only restricted the available post-mortem donor pool significantly, but the presence of cellular remnants in the ECM evokes immunological reactions from the recipient, causing tissue degeneration and graft failure [36]. Mitchell et al. (1998) reported the early death and loss of endothelial and interstitial cells in cryopreserved homografts following implantation, arguing that valve durability primarily relies on the retention of structural integrity and the preservation of the ECM instead of cell viability [37]. Smit et al. has previously shown that the post-mortem ischaemic time prior to the cryopreservation of homografts can be extended safely to around 48 h [6].

In the current study, the morphology and mechanical properties of standard cryopreserved ovine pulmonary homografts were compared to decellularized pulmonary homografts and decellularized pulmonary homografts treated with EnCap^TM^ AC technology. Homografts were harvested and processed after 48 h cold post-mortem ischaemia.

The histological evaluation of the cryopreserved homografts with H&E demonstrated the presence of an endothelial layer and uniformly distributed interstitial cells, but the collagen scaffold appeared collapsed (Figure 4A and Figure 5A). However, a limitation of H&E staining is the inability to differentiate between morphologically intact cells and non-viable cells [38,39]. SEM demonstrated a confluent endothelial cell layer on the leaflet and wall tissue of cryopreserved homografts, even after 48 h of ischaemia (Figure 6A,B). These endothelial cells presented with prominent nuclei and collapsed extranuclear areas, indicative of non-viable cells [8]. Shenke–Layland and co-workers described the presence of limited collagen-containing structures in the ventricularis of cryopreserved valve leaflets after thawing, with significantly altered and deteriorated collagenous and elastic fiber structures as a result of crystal ice formation in the ECM during cryopreservation [40]. This corresponds with our findings on TEM, where the collagen in the cryopreserved group appeared disrupted and damaged, with interstitial cells and cellular remnants (Figure 7A,B). These cellular remnants that remain in the ECM can activate the immune response from the recipient to the homograft and, together with the damaged and ruptured collagen, will result in early valve degradation [36].

Removing all the cellular content and DNA material from a homograft valve through decellularization results in an implant that significantly reduces the immunologic response from the recipient [41]. Decellularized valve leaflets and wall tissue demonstrated complete acellularity with H&E staining (Figure 4D and Figure 5D); however, the decellularization of leaflets using SDS has been shown to result in a dense ECM network and small pore sizes, which might make the recellularization of the matrix with interstitial cells more difficult [42]. Our devised decellularization process for homografts relies on combining osmotic shock, as an initial step to induce cell lysis, with detergents (SDS, SDC, and TritonX-100) and the addition of Benzonase^®^. The initial osmotic shock step reduces the concentrations and time of exposure of the tissue to the detergents and the enzyme, thereby reducing the damage to the collagen fibers and the ECM [43,44,45]. Ethanol was used to remove lipids [46,47]. The decellularization method used in this study resulted in collagen fibers that were more loosely organized with larger spaces between the fibers (Figure 4D and Figure 5D). SEM confirmed the complete removal of the endothelial cells on the leaflet and wall tissue of the decellularized valves, with only exposed collagen layers and remnants of basal membrane remaining (Figure 6), while TEM also demonstrated the loosely arranged collagen network (Figure 7C,D). The decellularization method proved effective in achieving complete acellularity, and the larger interfibrillar spaces could be very beneficial for in vivo recellularization once implanted while maintaining tissue strength when compared to cryopreservation (Table 1).

Converse and co-workers used a multi-detergent and enzymatic washout decellularization protocol, and differential scanning calorimetry (DSC) showed that this method does not reduce the cross-linking of collagen and thereby retains the strength of the tissue. However, the decellularization protocol they used reduced the GAG content, with resultant increased extensibility and changes in the relaxation behavior of the pulmonary valve leaflets [44]. The method currently used led to a reduction of the GAG content when used for the decellularization of bovine pericardium [48]. The GAG content was not evaluated in the current study, and the effect of this decellularization method on the GAG content of the pulmonary homografts should be investigated. However, the decellularization process did not significantly affect the stiffness of the homograft leaflet and the wall tissue compared to the cryopreserved tissue (Table 1). Although the Young’s modulus of decellularized leaflets suggests more extensible leaflet tissue after decellularization, the lack of significance may be due to the small number of samples tested. The comparable strength and Young’s modulus of the decellularized leaflet and wall tissue to that of the cryopreserved tissue could be attributed to the dehydration effect of ethanol in the delipidation step.

Based on these results, the decellularization of homografts using our proprietary method combined with an enzyme could be used as an alternative processing technique for homografts. Decellularized valves are currently being used in clinical practice. CryoValve^®^ SG (CryoLife, Inc, Kennesaw, GA, USA) pulmonary human heart valves were some of the first decellularized homografts to be used clinically for RVOT reconstruction, and the results compare favourably with cryopreserved homografts [49]; however, improved haemodynamics were observed in the CryoValve^®^ SG group, after four years of implantation, possibly due to decreased antigenicity of these valves following decellularization [50]. A European group led by Haverich has implanted 131 decellularized pulmonary homografts since 2005. A 10-year follow-up study found that decellularized homografts had a 100% freedom from explantation at 10 years follow-up, compared to 84.2% freedom from explantation for cryopreserved homografts and 84.3% freedom from explantation for GA-fixed bovine jugular vein conduits. Additionally, the rate of freedom of infective endocarditis was 100% in patients receiving decellularized homografts [51]. Although the decellularized homografts used by this group provided low gradients in follow-up and exhibited adaptive growth [12], the careful observation of their latest data shows no statistical difference in terms of stenosis and regurgitation when compared to the conventional cryopreserved homografts [52]. An early and midterm study by a group led by da Costa reported that decellularized aortic allografts replacements, the first of their kind, had 100% freedom from reoperation of the graft at 3 years follow-up. The decellularized aortic valves retained structural integrity, low calcification rates, and adequate haemodynamics [53].

Due to the concerns that the decellularization process might negatively affect the strength of the scaffold, the additional fixation and stabilization of the decellularized homografts were investigated. The additional fixation and detoxification of the decellularized valves were done using GA as a fixative and PG for detoxifying the GA-fixed homograft (EnCap^TM^ AC technology) [22]. This process resulted in more compacted and dense collagen layers when compared to the leaflet and wall tissue in the cryopreserved and decellularized groups (Figure 4G, Figure 5G and Figure 7E,F), which might make the recellularization of the matrix with interstitial cells more difficult [42]. Using a juvenile ovine model, Botes et al. (2022) demonstrated that commercial EnCap^TM-^treated bovine pericardium (not decellularized) had less host cell infiltration than the bovine pericardium processed using our proprietary decellularization process. The EnCap^TM-^treated tissue showed a loss of collagen fiber integrity, causing the collagen to become densely compacted, contributing to insignificant recipient cell infiltration [54]. The current study showed that even after decellularization, GA fixation led to the collagen layers of the homograft becoming compacted and dense, which might affect cell infiltration. Umashankar et al. (2012) reported the total lack of host tissue incorporation in GA-fixed bovine pericardium, compared to excellent host fibroblast incorporation in decellularized bovine pericardium. The authors conclude that this could be due to GA-treated bovine pericardium being resistant to collagenase, an important enzyme in the body that is responsible for ECM remodeling [55]. The additional fixation and detoxification of decellularized homografts using EnCap^TM^ AC technology added additional cross-links and increased the Young’s modulus of leaflets, although not significantly. GA fixation has an effect on the hydration properties of fixated tissue, impairing its ability to rehydrate, which is hypothesized to result from the reaction of GA with hydrophilic amine groups in collagen, reducing the tissue’s hydrating capacity and increasing stiffness.

## 5. Conclusions

In conclusion, the confluent layers of endothelial cells were still present on the leaflet and wall tissue of the cryopreserved group even after 48 h cold ischaemia. Our proprietary decellularization protocol with the additional Benzonase^®^ proved to be effective in removing all endothelial cells, interstitial cells, and DNA from the homograft wall and leaflet tissue, without a significant reduction in strength. The removal of donor cells and nuclear material might reduce recipient immunogenicity of the valve. Cryopreservation disrupted and fractured the collagen fibrils, while decellularization created larger interfibrillar spaces in the collagen network, which might be advantageous for host cell repopulation after transplantation. Elastin appeared to be well preserved in all three groups after processing, with no visible calcific nodules in any of the groups. The additional fixation of the decellularized scaffold created dense collagen networks and increased the leaflet stiffness, which could negatively affect the biomechanical behavior of the homograft. Due to the increased stiffness of the leaflets after fixation, additional fixation with GA following the decellularization of homograft valves is not desirable, but further investigations in this regard might be required. Compared to cryopreservation, the use of our proprietary multi-detergent decellularization method combined with the use of an enzyme is a promising alternative for homograft preservation. The clinical performance of these decellularized homografts should be investigated in pre-clinical (animal-based) studies.

## Figures and Tables

**Figure 1 polymers-14-03036-f001:**
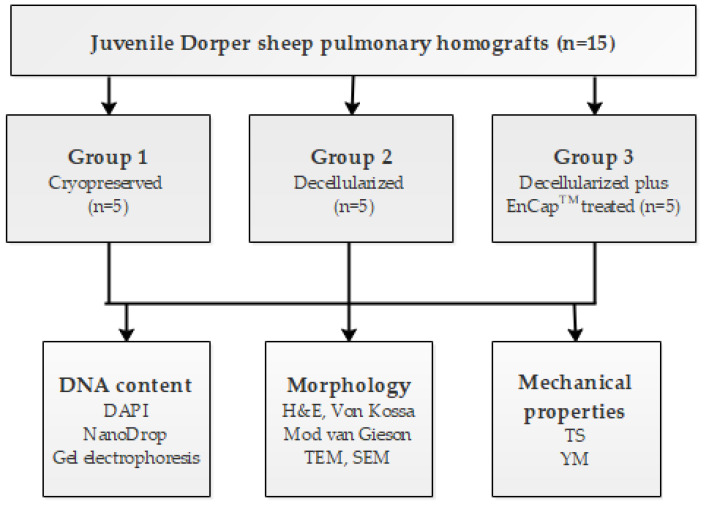
Schematic diagram of the study design.

**Figure 2 polymers-14-03036-f002:**
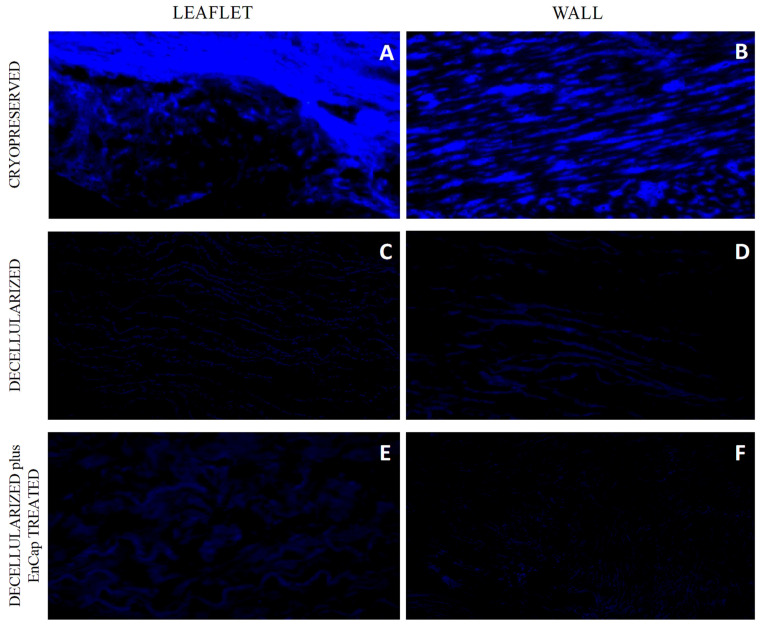
Representative images of DAPI-stained sections of cryopreserved (*n* = 5) (**A**,**B**), decellularized (*n* = 5) (**C**,**D**), and decellularized plus EnCap^TM^-treated (*n* = 5) (**E**,**F**) leaflet and wall tissue.

**Figure 3 polymers-14-03036-f003:**
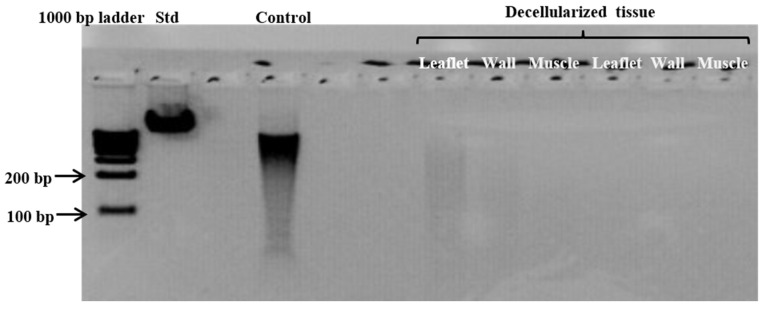
Representative gel electrophoresis image demonstrating the absence of DNA material in leaflet, wall, and muscle tissue of homografts after decellularization. (Standard (Std) = Lamba DNA; control = fresh homograft tissue).

**Figure 4 polymers-14-03036-f004:**
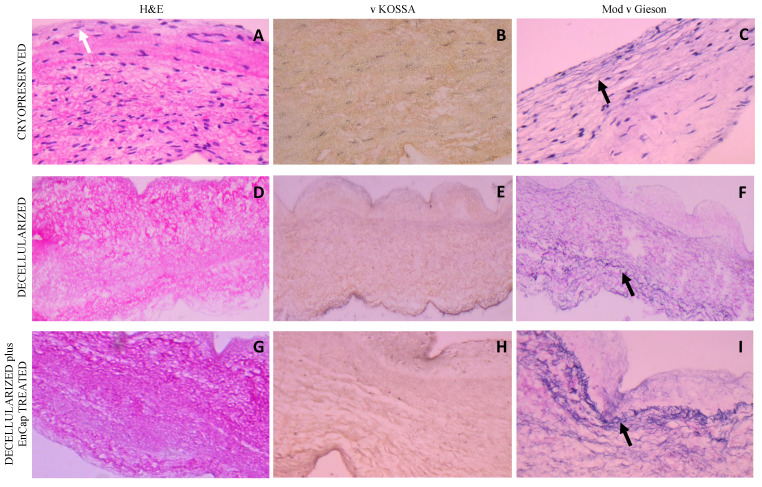
Representative H&E, von Kossa, and modified Verhoeff’s van Gieson staining of cryopreserved, decellularized, and decellularized and EnCap^TM^-treated leaflet samples processed after 48 h cold ischaemia; white arrow points to endothelial layer, and black arrows point to normally distributed elastic fibers ((**A**–**I**) = 400× magnification).

**Figure 5 polymers-14-03036-f005:**
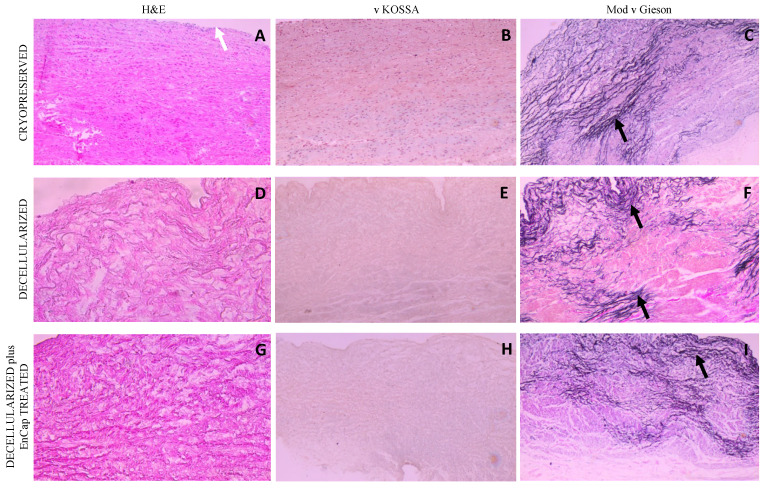
Representative H&E, von Kossa, and modified Verhoeff’s van Gieson staining of cryopreserved, decellularized, and decellularized and EnCap^TM^-treated wall tissue processed after 48 h cold ischemia; white arrow points to endothelial layer, and black arrows point to normally distributed elastic fibers ((**A**–**I**) = 100× magnification).

**Figure 6 polymers-14-03036-f006:**
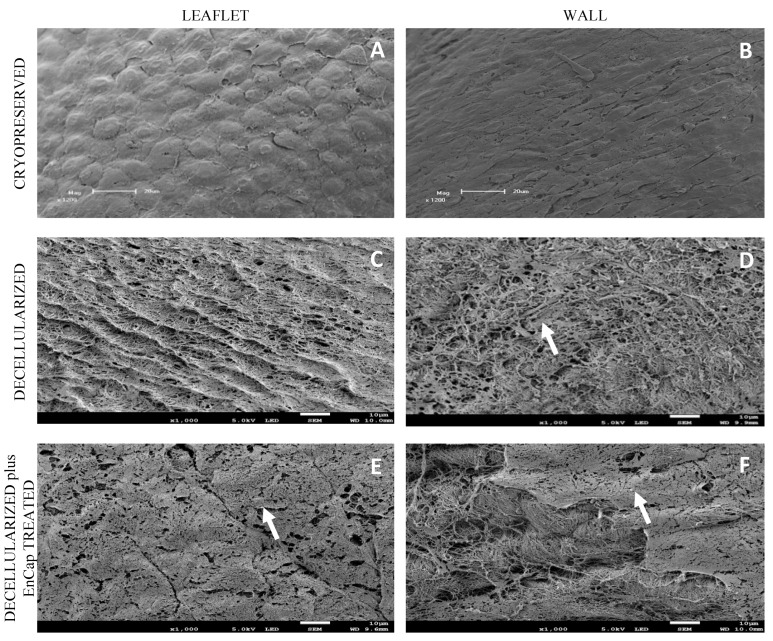
Representative scanning electron microscopy images of cryopreserved, decellularized, and decellularized and EnCap^TM^-treated wall tissue processed after 48 h ischemia; white arrows point to basal membrane remnants ((**A**,**B**) = 1200× magnification, scale bar 20 µm; and (**C**–**F**) = 1000× magnification, scale bar 10 µm).

**Figure 7 polymers-14-03036-f007:**
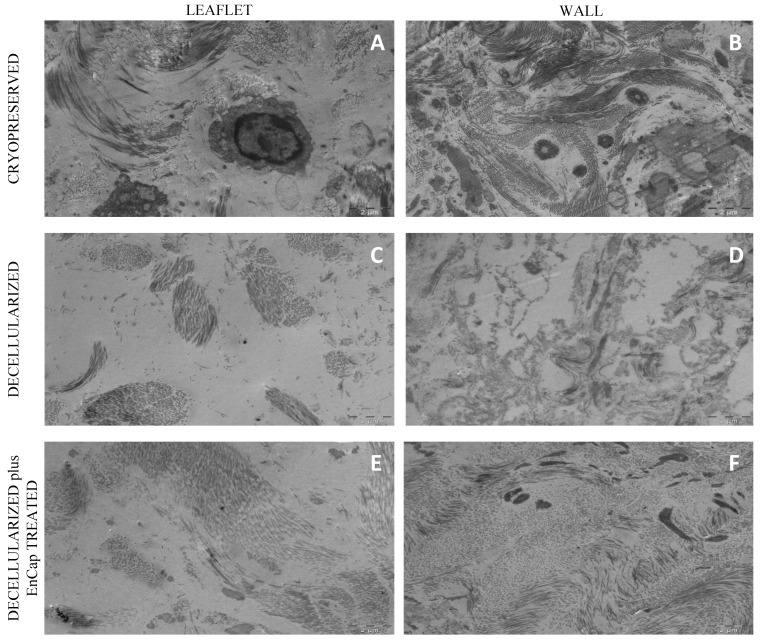
Representative transmission electron microscopy images of cryopreserved, decellularized, and decellularized and EnCap^TM^-treated wall tissue processed after 48 h ischemia ((**A**–**F**) = scale bar 2 µm).

**Table 1 polymers-14-03036-t001:** Baseline TS and YM of cryopreserved, decellularized unfixed, and decellularized plus EnCap^TM^-treated pulmonary homograft leaflets (*n* = 5) and wall tissue (*n* = 5).

Variable		Cryopreserved	Decellularized	Decellularized Plus EnCapTM-Treated	*p*-Values
**Leaflet**	TS (MPa)	3.72 (3.12–5.01)	1.83 (0.01–5.65)	4.96 (4.34–6.76)	0.23
	YM (MPa)	23.34 (14.12–28.87)	22.10 (0.10–55.27)	41.11 (27.76–52.52)	0.40
**Wall**	TS (MPa)	1.80 (1.35–2.65)	1.89 (1.57–2.38)	1.90 (1.18–2.09)	0.96
	YM (MPa)	1.97 (1.22–3.18)	2.32 (1.82–2.88)	2.21 (1.99–3.58)	0.75

Data presented as median (corresponding range (Q1–Q3)), the difference between groups is determined with the Kruskal–Wallis (KW) test, and no significant differences (*p* < 0.05) are indicated. (TS = tensile strength; YM = Young’s modulus).

## Data Availability

Data available from corresponding author on reasonable request.

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
