# Peer review of "Impact of Three Different Processing Techniques on the Strength and Structure of Juvenile Ovine Pulmonary Homografts"

_polymers, 2022, doi:10.3390/polym14153036_

Round 1
Reviewer 1 Report
The manuscript by van den Heever and cols aims to determine the effect of cryopreservation, and two protocols for decellularization on cellular content, structural integrity and mechanical properties of pulmonary homografts harvested from juvenile Dorper sheep. The authors demonstrate that whereas cryopreserved specimen retains some of its cells, their decellularization protocol successfully removes all traces of cells from the tissue. Differences in collagen structure were also demonstrated. Interestingly, a small, albeit not significant, difference in tensile strength and Young’s modulus was also observed.
1. This reviewer thanks the authors for the extensive background and review of the literature present in the introduction. Nonetheless, this section reads long and the goal of the manuscript and the important issue that it is looking to tackle gets somewhat lost in it. Perhaps the authors should consider moving part of it to the discussion and showcase the comparison between cryopreserved versus decellularized homografts, including the decellularization protocol previously described by the authors.
2. In addition to DNA, can the decellularization protocol also eliminate host cell proteins from the tissue?
3. Not that it is in the scope of the manuscript, but can the authors expand on how the changes in collagen structure between the decellularization and decellularization plus EnCap protocols may affect cell repopulation of the tissue?
Reviewer 2 Report
The manuscript is well structured and easy to follow. The three techniques studied are very well presented in the introduction, with their advantages and disadvantages, and also the authors use a large number of methods to investigate the strength and structure of the homografts. The purpose of the paper is also concisely stated.
However, a few minor changes are needed, namely:
- Figures 2 (C, D, E, F), compared to Figures 2 (A, B), are very dark, almost nothing can be seen in them. If the authors have this possibility, I recommend replacing them with brighter and clearer variants.
- I don't think it is necessary to repeat the explanations from the main text in the figure caption, as is the case with figures 4, 5 and 7, respectively.
For example, in Figure 4: lines 237-241 – the phrase “von Kossa staining showed no calcific… EnCap treated 240 group” is repeated in the main text between lines 227-233. Is it absolutely necessary?
Figure 5: lines 254-257 – “No calcific deposits were demonstrated…” is repeated between lines 245-250.
Figure 7: lines 276-278 - “Collagen bundles…” is repeated between lines 270-272.
I recommend deleting them, so that the figure captions appear more airy and clear.
- Conclusions: It should be a little more detailed. Only the effects on mechanical strength are mentioned. There is no mention of the histological evaluation; a brief conclusion would be needed in this case. Also, I recommend, a short comparison between the effects of the three techniques studied on the homographs, as a final conclusion.
